# Disentangling associations between pubertal development, healthy activity behaviors, and sex in adolescent social networks

Mark C. Pachucki[1,2]*, Lindsay Till Hoyt[3], Li Niu[3]ˮ, Richard Carbonaro[1]ˮ, Hsin Fei Tu[1]ˮ, John R. Sirard[4], Genevieve Chandler[5]

1 Department of Sociology, University of Massachusetts Amherst, Amherst, Massachusetts, United States of America, 2 UMass Computational Social Science Institute, University of Massachusetts Amherst, Amherst, Massachusetts, United States of America, 3 Department of Applied Developmental Psychology, Fordham University, Bronx, New York, United States of America, 4 School of Public Health & Health Sciences, University of Massachusetts Amherst, Amherst, Massachusetts, United States of America, 5 Elaine Marieb College of Nursing, University of Massachusetts Amherst, Amherst, Massachusetts, United States of America

ˮ These authors contributed equally to this work.
* mpachucki@umass.edu

**Data Availability Statement:** This research uses third party data from the National Longitudinal Study of Adolescent to Adult Health (Add Health).

## Abstract

With the onset of puberty, youth begin to choose their social environments and develop health-promoting habits, making it a vital period to study social and biological factors contextually. An important question is how pubertal development and behaviors such as physical activity and sleep may be differentially linked with youths' friendships. Cross-sectional statistical network models that account for interpersonal dependence were used to estimate associations between three measures of pubertal development and youth friendships at two large US schools drawn from the National Longitudinal Study of Adolescent to Adult Health. Whole-network models suggest that friendships are more likely between youth with similar levels of pubertal development, physical activity, and sleep. Sex-stratified models suggest that girls' friendships are more likely given a similar age at menarche. Attention to similar pubertal timing within friendship groups may offer inclusive opportunities for tailored developmental puberty education in ways that reduce stigma and improve health behaviors.

## Introduction

Certain stages of the life-course can have disproportionate influences on future social development and health [1–3]. With the onset of puberty and adolescence, youth begin to choose their own social relationships, environments, and develop activity patterns [4], making it a vital period to study how social and biological factors are intertwined. During this time, scholars have noted that certain key physical behaviors change significantly: physical activity levels decline, and older adolescents go to bed later and sleep less than younger adolescents in ways that vary by sex and race/ethnicity [5–8]. Adequate physical activity has decreased for boys and remains low for girls [9], and adolescent sleep has declined considerably from 1991 to 2012 [10]. Importantly, it has also been established that during adolescence, patterns of social

Information on how to obtain the Add Health data files is available on the Add Health website (http://www.cpc.unc.edu/addhealth). To apply for these data, the researcher must request access and download a contract application from the UNC Carolina Population Center Data Portal (https://data.cpc.unc.edu/projects/2/view). Once completed, researchers must submit their application through the portal. Data used in the current study were not collected by the study authors and therefore we do not have the permission to share the data. The data used are restricted use data from the Add Health study. Researchers who wish to gain access to this data must apply for access using the information specified above. The authors did not have any special access privileges to the data and confirm that the data was obtained in the same manner outlined above. Relevant codes for replicating analyses are within the Supporting information files of this manuscript.

**Funding:** Research reported in this study was supported by the National Institute of Nursing Research under Award Number R21NR017154 to Co-PIs Mark C. Pachucki and Lindsay T. Hoyt. This research uses data from the National Longitudinal Study of Adolescent to Adult Health (Add Health). Add Health is directed by Robert A. Hummer and funded by the National Institute on Aging cooperative agreements U01 AG071448 (Hummer) and U01AG071450 (Allison E. Aiello and Hummer) at the University of North Carolina at Chapel Hill. Waves I-V data are from the Add Health Program Project, grant P01 HD319121 (Harris) was directed by Kathleen Mullan Harris and designed by J. Richard Udry, Peter S. Bearman, and Kathleen Mullan Harris at the University of North Carolina at Chapel Hill, and funded by Grant P01-HD31921 from the Eunice Kennedy Shriver National Institute of Child Health and Human Development, with cooperative funding from 23 other federal agencies and foundations. No direct support was received from grant P01-HD31921 for this analysis. The funders had no role in study design, data collection and analysis, decision to publish, or preparation of the manuscript.

**Competing interests:** The authors have declared that no competing interests exist.

relationships are associated with these same activity behaviors [11–15]. Pubertal development is also associated with activity behaviors [16–18], with some evidence of pubertal development being correlated among friends [19, 20].

Yet it remains an outstanding question how pubertal development and these related physical behaviors (physical activity and sleep) may be differentially associated with youth socialization, because these factors tend to be studied in isolation. Why might puberty and activity behaviors be associated with an adolescent's friendship networks, and why should we pay attention to these associations? One response to these questions has to do with homophilic selection processes–the idea that youth may choose to become friends with others who enjoy similar activities, come from similar backgrounds, or have similar development status. Network analysis is positioned to address this analytic challenge of disentangling these selection processes by modeling the complex patterns of relationship dependence between individuals [21, 22]. As such, the goals of this study are to: (1) determine the extent to which pubertal development and activity behaviors (physical activity and sleep) may independently be associated with social relationships among friends using exponential random graph modeling, a form of network analysis which is able to account for correlated observations between individuals, and (2) to discern how these associations may vary by sex. The idea that pubertal development and activity behaviors may be patterned within adolescent friendship networks is supported by a small but growing body of evidence from the landmark U.S.-based National Longitudinal Study of Adolescent to Adult Health (Add Health) study [23], as well as international cohort studies [19, 24]. Knowing of similar pubertal timing within friendship groups may offer inclusive opportunities for tailored developmental puberty education in ways that reduce stigma by normalizing conversations about physical and physiological changes during adolescence with trusted adults.

## Puberty and friendship group dynamics

Puberty is often a time of mismatch in social and biological development. During this period of transition, emotional and cognitive maturity can be temporarily out of sync with behavioral expectations for individuals of a given age. Being off-time from peers is thought to be stressful, as newfound physical distinction can elicit different reactions from both adults and peers. Maturing earlier than peers (i.e. early timing) is theorized to be associated with an abbreviated timespan in which to develop a strong sense of self-identity and find accepting friends [25, 26]. In the case that puberty begins when expected, youth can anticipate, prepare, and learn how to cope with their changing situation in community with their peers, whereas youth who begin developing before (or after) their peers may face insecurity and heightened stress.

In one of the only large-scale studies to date to directly focus on the network distribution of pubertal development within peer friendship groups [20], Kretsch and colleagues explored multiple measures of development: one measure based upon an adolescent's perception of themselves relative to peers (i.e., relative pubertal timing), and another of stage-normative physical development based upon more traditional physical indicators of puberty (i.e. in boys, voice change and body hair; in girls, breast development, menarche, body curviness). The investigators examined a cross-section of more than 2800 participants (middle to late adolescence) in the Add Health sample during 1994–95, using partial correlation analysis of pubertal development measures pooled across 16 schools, intra-class correlations to assess between-school variation in these measures, and sex-stratified mixed-effects models predicting different pubertal timing measures to account for between-school pubertal timing clustering. The authors found that for boys, stage-normative pubertal development was positively associated with that of their nominated (male) friends, and that for girls, relative pubertal timing was

similar to nominated (female) friends. The authors also examined whether this apparent clustering differed by school, finding between-school differences in boys' stage-normative pubertal development after adjusting for mean school age and percentages of Black and Hispanic students at each school, given racial/ethnic differences in pubertal onset.

Several takeaways are that even with relatively rough network measures of peer pubertal development (relying upon an average of nominated friends' development measures), the authors document associations within friend groups. These associations vary by sex, and importantly, these associations also differ by local school environment. However, the design of this study only focused on same-sex friendships, precluding analysis of pubertal development and cross-sex social ties. Perhaps more importantly, what this or any type of analysis conducted in a regression framework *cannot* do is account for the statistical dependencies between youth–that two individuals nominated as friends by a given youth may themselves be connected as friends. These types of endogenous network properties are critical to consider.

A separate body of work has shown that variation in pubertal development is also related to risky behaviors, such as various forms of delinquency and age of sexual debut, which are also associated with friendship dynamics [27, 28]. Most relevant to the questions motivating this study, Franken et al. [19] performed a statistically rigorous longitudinal network-behavior study of friendship formation and externalizing behaviors (delinquency, smoking, drinking) among early adolescents in two different schools in the Netherlands over the course of a year ($\mu$ = 12.7 years of age, equivalent to 7th-8th grade in the US). The study design was able to account for endogenous network properties in contrast to previous regression-based research. The authors found that youth who had similar physical development were more likely to form friendships with one another, and that physically more advanced were more likely to retain friendships with students who engaged in similar externalizing behaviors. A separate longitudinal three year network-behavior study of Taiwanese youths' substance use and network change (grades 7 to 9, age $\mu$ = 12.3) gathered information on the Pubertal Development Scale to ascertain whether youth were on-time, early, or late relative to peers [24]. The authors did not observe either selection or network influence effects on pubertal development. However, whether this tendency towards friendship among similar-development youth applies in the United States (US) context, and whether it holds for physical behaviors (vs. externalizing risk behaviors) remain open questions. Speculatively, given sociocultural norms between the Netherlands (more Western social context) and Taiwan (more collectivistic), we might expect a closer fit with findings from the Dutch youth study.

## Physical behaviors and networks

While some prior research concerning pubertal development and health behaviors focuses on risk behaviors related to delinquency, drug use, and sexual activity, another subset of research treats youths' physical behaviors as different types of risk critical to cardiometabolic health [29–32]. Network research has shown that adolescent physical activity (PA) and sleep behaviors are correlated within youth social networks [for reviews, see 12, 33, 34]. For instance, among a cohort of Australian early adolescents (roughly ages 13 to 14), cross-sectional exponential graph models at two different schools revealed that same-sex friends were similar in terms of organized physical activity behaviors [35]. Sirard and colleagues [36] investigated moderate-to-vigorous PA among nearly 2,800 students from 20 public schools in Minneapolis & St. Paul, Minnesota, finding that PA levels were similar among socially-connected friends, with slightly stronger associations for girls.

More rigorous longitudinal network research designs, however, have attempted to discern reasons for similarity in physical activity (e.g. selection or social influence mechanisms).

Drawing upon participants in two of the largest schools in the Add Health parent study ('Sunshine High', a pseudonym for a suburban public high school in the western US; and 'Jefferson High', a public high school in the rural Midwest), Shoham and colleagues found that in both schools, adolescents selected one another, in part, based on the volume of active sports they engaged in [37]. There was also evidence of social influence on changes in active sport levels. A separate team advanced an elaborated model specification by simultaneously modeling physical activity and weight change as behaviors that jointly co-evolve with friendship formation, including different weight-related confounders and interactions in models. This study found that at both schools, adolescents select one another for friendships partially because of shared PA levels and body-mass index (BMI), but that ties to friends also influence changes in PA and BMI over time [13].

While studies documenting causes for adolescent sleep decline are clear that electronically-mediated interactions with friends are associated [38, 39], there has been a limited amount of network research on sleep. Two exceptions are a longitudinal study based upon more than 8,000 participants in Add Health at Waves I & II, in which the authors used generalized estimating equation regression models and found an increased risk of poor sleep if one's friends previously reported sleeping poorly, adjusting for one's own baseline sleep and other covariates [15]. A more recent Add Health study restricted to students in Sunshine High and Jefferson High (n = 2,550 participants) used linear regression analyses to find that sleep insufficiency was associated with network popularity (a greater amount of friendship nominations) among girls [40]. Finally, one recent study relying upon data from Waves I-IV of Add Health investigated the relationships between pubertal timing and risk/health promotion behaviors, including sleep health, in the full sample. Though this study did not examine social networks, earlier pubertal timing put girls at risk for less sleep during adolescence. However, there was also some evidence that later timing is associated with health promoting (e.g., more sleep) activity behaviors [41]. Specifically, for both boys and girls, later adolescent physical development was prospectively associated with reporting more sleep nightly as young adults.

## Hypotheses

To briefly review, prior research on puberty and networks has suggested some correlation among friends' pubertal development using cross-sectional regression-based methods: one study in a US secondary school context (Add Health) pooled respondents across more than a dozen schools, while two separate studies based on a Netherlands sample and Taiwan sample of youth, respectively, both used stochastic actor-oriented models, finding in the Dutch case that youth selected one another for friendships, in part, based on pubertal development. Separately, prior actor-oriented research has demonstrated evidence of both social influence and friendship selection mechanisms associated with adolescents' physical activity among students in Add Health, and prior regression-based studies have shown sleep to be associated with friendship patterns as well.

While these studies have been illuminating, they were also not designed to provide answers to the central question of this study: How are pubertal development and important physical behaviors associated with an adolescent's patterns of friendships? On one hand, the prior use of regression-based methods for US adolescent data pose difficulties in accounting for network dependencies among youth who know one another and the associated endogenous network processes that play into friendship formation. On the other hand, prior network studies that account for statistical dependence and that integrate puberty measures into analyses have, to date, only been conducted in countries outside the US. Thus, an appropriate starting point for analysis of our research question is to use a statistical network approach, exponential random

graph models (ERGMs), that can appropriately handle dependence concerns. While we initially considered a longitudinal network-behavior framework such as SAOMs (and indeed, some earlier Add Health network studies have used SAOMs), we felt that given our relatively narrow research question it would be more conservative to use ERGMs, especially given that model developers have recommended the use of more than two waves of network behavior data for best model fit [42].

Given prior research with the Add Health cohort, it is important to first conceptually replicate the same patterns of PA and sleep homophily among friends using an ERGM framework that others have observed using different methods. This will enable better contextualization of results with earlier efforts. Beyond this, however, we test two novel hypotheses, first, that adolescents' pubertal development will be associated with a greater likelihood of friendship formation with them (Hypothesis 1). Prior network research does not lead us to clearly predict a direction for this association, though developmental research gives some guidance. On one hand, it is plausible that youth who report being significantly *earlier* developmentally than peers will have a lower odds of friendship formation due to the stress of being more physically advanced than one's peers–being more physically developed than peers can be a social asset though also stigmatizing. Alternatively, being significantly *later* than peers might lead individuals to simply join younger (or younger-looking) friendship groups, though this seems less likely given how school grades are largely age-tracked. On balance, it is likely that being more developed–even if earlier than peers–will be associated with more friendships.

Further, prior research on concordance in pubertal status among nominated friends [20] suggests that pubertal development among socially-tied friends will also vary by sex (Hypothesis 2), though, given lack of prior network research in this area, it is difficult to predict whether specific measures of pubertal development will be associated with social ties among boys or girls. Given that girls develop earlier than boys, it is likely that we will observe these associations more clearly for girls. Last, given that individual schools have different sizes, demographic compositions, climate and cultures, and organizational differentiation–what McFarland and colleagues [43] describe as 'network ecology'–we do not expect a consistent magnitude of effect sizes between schools, though we do have a minimum expectation of a similar direction for associations between pubertal development and friendship ties.

H1: Adolescents' pubertal development will be associated with a greater likelihood of friendship formation with them.

H2: Pubertal development among socially-tied friends will vary by sex, with pubertal timing homophily more apparent among girls than boys.

## Materials and methods

To test these hypotheses, analyses draw from the initial wave (1994–95) of the National Longitudinal Study of Adolescent to Adult Health (Add Health). In the original parent study, high schools (n = 80) and feeder middle schools (n = 52) were selected from a stratified nationally representative sample of all public and private schools in the U.S. In-home interviews were conducted with 20,745 students and families. From these schools, 16 were selected for inspection (commonly called "saturation schools", because they represented an above-average attempt at increased recruitment by study investigators and have more complete data). Given the range of confounders that were necessary to account for, we chose to focus on the two largest of these sixteen schools. These schools are referred to in prior research by their pseudonyms, "Sunshine High" (n = 1721), and "Jefferson High" (n = 832). After excluding participants who did not have valid grade-level information, the sample size decreased slightly

(Sunshine n = 1658; Jefferson n = 806). After excluding participants who did not have complete covariate data, the sample size decreased further (Sunshine n = 1568; Jefferson n = 775). Missing covariate proportions at Sunshine High were parent education (3.4%), relative puberty (1.15%), physical development (1.6%), race (0.54%), ethnicity (0.36%), screen time (0.18%), sleep duration (0.18%). sleep sufficiency (0.18%). Missing at Jefferson High were parent education (1.6%), relative puberty (1.12%), physical development (1.6%), screen time (0.12%), sleep duration (0.12%).

After excluding participants who did not make any valid in-school friend nominations, we arrived at the final analytic sample (Sunshine n = 1408; Jefferson n = 762). We did not impute missing data due to strong assumptions required to do so in a network context, where data may be missing on both individuals' attributes as well as missing on possible social ties between them [44]. While research on network imputation has advanced in recent years [45–47] the complexities of model-based imputation strategies are not yet easily implemented on large and sparse (decentralized) networks such as the ones we focus on in this study. This study used de-identified secondary data from human participants, accessed March 2023, in accordance with the ethical standards of the University of Massachusetts under human subjects protocol #1059.

### Measures

**Friendship relations.**   Social ties (Wave I) were identified by asking participants to name up to 10 friends (5 male, 5 female) during interviews. Crucially, because nominated friends were also study participants, their respective pubertal development, physical behaviors, and other attributes are informative in analyses. We constructed networks based upon directed social tie information, but because of the difficulties of estimating large models using directed ties, chose to symmetrize networks such that a tie linking participant $i \longrightarrow j$ or participant $j \longrightarrow i$ results in an undirected tie between $i$ and $j$. Nominations to friends outside a given school were discarded.

**Pubertal development.**   Three measures of puberty were assessed. The first was a measure of physical development derived from selected questions about intra-individual changes based on the Pubertal Development Scale (PDS), the most widely used self-report [48]. These questions focus on bodily changes that become evident in mid-to-late puberty (e.g. body hair for boys, breast development for girls). Respondents were asked to answer about these bodily changes relative to "when I was in grade school" as an anchor. We then coded responses to these measures into a continuous scale that ranged from 1 (least developed/very little change) to 5 (most developed/a lot of change) by taking the mean of responses to three questions for boys (i.e., facial hair, body hair, and voice change) and two questions for girls (i.e., breast development and body curviness) and combining them into a single scale. A second measure of puberty was perceived inter-individual development, or peer-relative pubertal timing. This measure was assessed with the question, *"How advanced is your physical development compared to other boys/girls your age?"* Responses could range from "1: I look younger than most", "2: I look younger than some ", "3: I look about average", "4: I look older than some", and "5: I look older than most". The third measure was age of menarche, for girls, and across the schools it ranged from 7 to 16 years. Although menarche was also included in the original PDS, separating it from our measure of physical development allows for the remaining items to contribute to a common development measure across boys and girls, and allows for evaluating menarche as a distinct contributor to friendship in girls-only models.

**Physical behaviors.**   Self-reported physical activity (PA) was assessed by summing questions on activity type and frequency. Previous work has used these questions to measure

environmental determinants of PA [49], adolescent to adult behavior patterns [50, 51] associations between future expectations and health behaviors [52], and the link between family relationships and PA [53]. In terms of PA, following exemplars [13, 52], we summed the three available measures of past-week activity to provide a holistic measure: the frequency of skating/biking, the frequency of playing an active sport, and the frequency of exercising.

Several Add Health questions have been used to measure sleep, including a question about typical sleep duration (hours) and whether or not participants felt they got sufficient sleep in an average night [7, 54]. We use both sleep duration and sleep sufficiency measures because of individual differences in need for sleep.

**Confounders.** Information was gathered on participant sex (Male/Female–no gender identity information was available); age (based on date of birth); race (White; Black; Asian American /Pacific Islander, AAPI; Native American; Other race); Hispanic ethnicity; and highest parent education (<HS, HS graduate/GED, some college, college graduate, post-college education). In addition, models adjust for participant grade. Because pubertal development, activity behaviors, and socialization are often related to weight status, we also include an indicator for "overweight or obese" derived from self-report of height and weight using Centers for Disease Control sex and age-specific thresholds for children implemented in the Stata 'zanthro' package [55]. To measure screen time as a confounder of primary physical behaviors (PA and sleep), we summed three measures: TV hours (per week), video hours, and screen games [49]. How youth make use of screens in the modern era (screens as a two-way medium) is drastically different than how youth in the mid-1990s made use of screens (a one-way medium). While we would have liked to analyze youth screen time as a primary activity behavior (given its links to both physical activity and sleep), we were convinced by helpful reviewers that doing so would primarily be of historical interest rather than generalizable to today's youth. Last, because of a programming irregularity in the parent study, not all participants were afforded a chance to nominate the full list of 5 female and 5 male friends. Those who were restricted to one best friend of either sex were denoted with an indicator variable ("restricted nomination"); this affected approximately 4% of Sunshine participants, and 5% of Jefferson participants.

## Considerations regarding modern adolescent socialization, pubertal development, and activities

It is important to note that *how* adolescents socialize in today's smartphone era differs considerably from the 1990s, and today, offline and online interactions need to be considered carefully [56–58]. Screens are now ubiquitous in a way that is qualitatively different from youth experiences with visual media in the 1990s. Yet because these friendship data were collected nearly 30 years ago, they enjoy the now-rare advantage of avoiding measurement complications between online and offline ties. Today's youth are broadly similar in many ways as well. Then, as now, physical proximity, or propinquity, remains a dominant driver of friendship in schools. Physical activity measures, while lacking the precision of contemporary measures like GPS-enabled accelerometry, are still quite informative about variation in common youth activities. While greater precision of sleep measurement is also now possible with wearable monitors, self-reported sleep sufficiency and sleep duration measures are still widely used. In terms of pubertal development, the onset of puberty has rapidly declined since the industrial revolution [59] and has continued to decline (albeit more slowly) during the past several decades [60, 61]. Thus, if, as we hypothesize, similarity in pubertal development and physical behaviors are associated with greater likelihood of friendship ties, we would predict such estimates to represent a relatively conservative snapshot of a patterned social phenomenon that is likely to be

even more stark today. In sum, we take the position that these analyses can be informative to the current era and not simply historical analysis, a point we return to in our conclusions.

## Analytic approach

Social network analysis (SNA) is a field of inquiry that seeks to explain the causes and consequences of patterned social relationships among humans; a wealth of useful guides are readily accessible [e.g. 62–64]. A key feature of SNA is a practical and statistical concern not just for individual attributes and qualities, but also a concern with quantities and qualities of ties between individuals and how–together–individuals and their social ties are connected with a given phenomenon. One type of SNA, exponential random graph models (ERGMs), were used to test study hypotheses and estimate determinants of friendship formation separately within each school using the software package R [65]. Briefly, ERGMs are simulation-based statistical models designed to estimate the patterns of tie formation and dyadic dependencies in relational data [66, 67]. In this model framework, the outcome of interest is the probability that an adolescent will become a friend with another adolescent. In this framework, ERGMs predict the probability of tie formation incorporating both network attributes and covariates pertaining to adolescents as well as explicitly accounting for interpersonal ties among them. We use ERGMs to model the network as a function of network structure, individual and dyadic covariates [65, 68] as follows:

$$\Pr(\mathbf{Y} = \mathbf{y}) = \left( \frac{1}{\kappa} \right) exp \left\{ \sum_A \eta A g A(\mathbf{y}) \right\} \tag{1}$$

In Eq 1, the probability of tie formation with another youth (Y) is conditional on the matrix of covariates, while $\eta A$ are coefficients of covariates and $g A$ are network statistics [65, 69, 70]. $\kappa$ represents a normalizing quantity for the distribution [65].

There are three types of covariates included in models: structural (endogenous) network effects, individual-level covariates, and dyadic covariates, and with limited exceptions (race and ethnicity covariates, discussed below), we sought to estimate the same model specification across both schools. For structural network effects, we adjust for the general rate of tie formation ("edges") in the network, for geometrically-weighted degree in the network (a measure of an individual's network size), and for geometrically-weighted edgewise shared partners (a measure of triadic closure). Individual covariates include three measures of pubertal development (physical development, peer-relative timing, and age of menarche for girls), physical behaviors (weekly PA, sleep duration, and sleep sufficiency), sex, school grade, age, parent's highest education level, and race and ethnicity (at Sunshine only, as there was minimal variation at Jefferson).

The dyadic covariates in this paper measure different forms of homophily in the network, the tendency for ties to form between two linked adolescents with the same attributes. Here, we measure same-grade (within each grade), same-race (within each category), same-ethnicity, same-physical development, same-peer-relative-timing, same-PA, same-sleep duration, and same-sleep sufficiency. For continuous variables (PA, sleep duration, physical development, peer-relative timing, menarche age for girls), we use a term to measure the absolute difference in values between any two individuals on that covariate [71]. A negative coefficient on continuous similarity ("absdiff") indicates stronger homophily (since absolute differences in a value decrease as values trend towards being more similar). The remaining dyadic similarity variables are dichotomous (1 = same, 0 = different), and use a "nodematch" term to calculate categorical similarity (a positive coefficient indicates greater friendship likelihood, a negative coefficient lower). To address questions about sex differences in determinants of tie formation,

we take a network stratification approach, following de la Haye and colleagues [35] and generate a male-only network and female-only network in each school, as they did. To exhaustively capture all ties, we also construct a residual subnetwork which is solely comprised of cross-sex social ties (male-female).

## Results

Table 1 summarizes the sample characteristics of participants at both schools. The broad contours of the social and demographic characteristics of students at these schools are well-known, and given that the analytic sample is largely similar to other published work [13, 20, 40] we do not go into great descriptive detail. However, with regard to both pubertal development measures, participants at Jefferson High report being slightly more developmentally mature than those at Sunshine (both in terms of physical development, as well as perceptions of pubertal development relative to same-sex peers), which is curious given the younger average age of Jefferson participants ($\mu$ = 16.9 and SD = 1.2, relative to $\mu$ = 17.3 and SD = 1.0 at Sunshine). It is already known that given the smaller school size of Jefferson and greater network density, participants report a greater number of friends than at Sunshine. Jefferson students tend to have slightly healthier behaviors (more PA, a greater amount of sleep and sleep sufficiency, less screen time), and are less overweight/obese.

### Correlates of friendship formation

Table 2 reports on estimates of stepwise models across both schools. On the whole, model diagnostics indicated adequate VIF metrics, adequate degree parameters, suboptimal edgewise shared partners at segments of the distribution, relatively poor geodesic distance (expected given large schools) and appropriate MCMC algorithm convergence, though Sunshine was on the whole better-fit than Jefferson. GOF and MCMC plots for both schools' models are available in the S1 File. The base model includes key endogenous network terms, the second adds puberty predictors, the third adds activity behaviors and confounders, and the last adds dyadic homophily terms. At both schools, as others have shown, there is a tendency towards larger network size being associated with tie formation (greater degree), and a higher likelihood of friendship based upon having friends in common (geometrically-weighted edgewise shared partners). In addition, tendencies towards same-race, same-sex, and same-grade social ties among youth have been well-established among youth across multiple schools [66]. Youth at both schools with more PA tend to have more friendships, and at Sunshine, youth reporting longer sleep tend to have fewer friendships. There is also a tendency for older and higher-SES youth to form more friendships. At Sunshine High, where there is greater racial and ethnic diversity, Black and AAPI youth are less likely to form friendships relative to White non-Hispanic students, while those classified as 'Other non-Hispanic' race are more likely to form friendships.

The set of dyadic terms included in the model indicate a tendency towards same-sex, same-grade, and same-weight status friendships (at both schools), and same-race and same-ethnicity friendships (at Sunshine High, which has sufficient racial diversity to test for this association). In terms of the key dyadic terms of interest (pubertal development and activity behaviors), youth with similar PA levels at both schools tended to form ties (Sunshine High, b = -0.024, p = <0.001, Jefferson High, b = -0.020, p = 0.0002). Youth at Sunshine High with similar relative pubertal timing (b = -0.040, p = 0.032) tended to be friends. Youth at Jefferson who had similar sleep duration also tended to be friends (b = -0.032, p = 0.02). Again, for interpretation purposes, we reiterate that negative dyadic homophily coefficients indicate *greater* similarity because the model term is based upon absolute difference. Having the same sleep sufficiency

status was not significantly associated with tie formation at either school below p<0.05, though at both schools there are indications of a trend in this direction of similar size. Because interpreting unadjusted coefficients in ERGMs is not necessarily straightforward for readers unfamiliar with this statistical approach, we contextualize that the magnitude of these dyadic

**Table 1. Participant characteristics (Baseline, 1994–95).**

| Characteristics | School 1 ("Sunshine High") | | | | | School 2 ("Jefferson High") | | | | |
|---|---|---|---|---|---|---|---|---|---|---|
| | n = | μ / % | SD | Range | | n = | μ / % | SD | Range | |
| *Pubertal timing* | | | | | | | | | | |
| Physical development | 1,408 | 2.70 | 0.97 | 1 - | 5 | 762 | 3.07 | 0.88 | 1 - | 5 |
| Peer-relative pubertal timing | 1,408 | 2.97 | 1.09 | 1 - | 5 | 762 | 3.36 | 1.05 | 1 - | 5 |
| Age at menarche (Girls only) | 632 | 12.21 | 1.39 | 8 - | 16 | 343 | 12.29 | 1.32 | 7 - | 16 |
| *Activity Behaviors* | | | | | | | | | | |
| Physical activity, times/wk | 1,408 | 6.24 | 3.96 | 0 - | 18 | 762 | 6.78 | 4.19 | 0 - | 18 |
| Sleep duration, avg hours/wk | 1408 | 7.35 | 1.26 | 3 - | 12 | 762 | 7.59 | 1.39 | 4 - | 14 |
| Sufficient sleep | | | | | | | | | | |
| No | 524 | 37.2% | | | | 232 | 30.4% | | | |
| Yes | 884 | 62.8% | | | | 530 | 69.6% | | | |
| *Socio-demographic covariates* | | | | | | | | | | |
| Gender | | | | | | | | | | |
| Female | 679 | 48.2% | | | | 352 | 46.2% | | | |
| Male | 729 | 51.8% | | | | 410 | 53.8% | | | |
| Grade | | | | | | | | | | |
| 9 | - | - | | | | 223 | 29.3% | | | |
| 10 | 489 | 34.7% | | | | 218 | 28.6% | | | |
| 11 | 494 | 35.1% | | | | 174 | 22.8% | | | |
| 12 | 425 | 30.2% | | | | 147 | 19.3% | | | |
| Age | 1408 | 17.3 | 1.0 | 15 - | 21 | 762 | 16.9 | 1.2 | 15 - | 20 |
| Race | | | | | | | | | | |
| White | 283 | 20.1% | | | | 742 | 97.4% | | | |
| Black | 321 | 22.8% | | | | - | - | | | |
| Asian | 477 | 33.9% | | | | - | - | | | |
| Other | 327 | 23.2% | | | | 20 | 2.6% | | | |
| Hispanic | 555 | 39.4% | | | | - | - | | | |
| Parent's highest education | | | | | | | | | | |
| <HS grad | 337 | 23.9% | | | | 46 | 6.0% | | | |
| HS grad/GED | 325 | 23.1% | | | | 296 | 38.8% | | | |
| Some college | 286 | 20.3% | | | | 183 | 24.0% | | | |
| College grad | 338 | 24.0% | | | | 168 | 22.0% | | | |
| Post-college | 122 | 8.7% | | | | 69 | 9.1% | | | |
| Overweight/Obese | | | | | | | | | | |
| No | 982 | 69.7% | | | | 578 | 75.9% | | | |
| Yes | 426 | 30.3% | | | | 184 | 24.1% | | | |
| Screen time scale, hours/wk | 1,408 | 22.10 | 17.99 | 0 - | 115 | 762 | 18.77 | 18.67 | 0 - | 175 |
| Restricted friend nominations | | | | | | | | | | |
| No | 1,355 | 96.2% | | | | 723 | 94.9% | | | |
| Yes | 53 | 3.8% | | | | 39 | 5.1% | | | |
| *Network attributes* | | | | | | | | | | |
| Degree (mean) | 1,408 | 3.86 | 2.63 | | | 762 | 6.62 | 3.54 | | |
| Edges | 2717 | | | | | 2521 | | | | |

**Table 2. Tie formation models, two large schools.**

| | School 1 ("Sunshine High") | | | School 2 ("Jefferson High") | | |
|---|---|---|---|---|---|---|
| | **Puberty** | **+ Confounds** | **+ Dyadic terms** | **Puberty** | **+ Confounds** | **+ Dyadic terms** |
| Edges | -6.858*** | -5.001*** | -6.327*** | -5.897*** | -4.888*** | -6.567*** |
| | (0.096) | (0.504) | (0.900) | (0.094) | (0.367) | (0.563) |
| GW Degree (0.2) [a] | 1.057*** | 1.404*** | 1.494*** | 0.433 | 0.614* | 0.610* |
| | (0.154) | (0.169) | (0.171) | (0.282) | (0.284) | (0.290) |
| GW ESP (0.2) [a] | 1.948*** | 1.930*** | 1.669*** | 1.681*** | 1.670*** | 1.536*** |
| | (0.031) | (0.031) | (0.032) | (0.036) | (0.035) | (0.036) |
| Pub: Physical dev | 0.011 | 0.017 | 0.017 | 0.001 | 0.008 | 0.013 |
| | (0.012) | (0.012) | (0.012) | (0.013) | (0.013) | (0.013) |
| Pub: Relative timing | 0.002 | 0.004 | 0.005 | -0.005 | -0.006 | -0.009 |
| | (0.010) | (0.011) | (0.011) | (0.010) | (0.010) | (0.010) |
| Physical activity, times/week | | 0.008** | 0.011*** | | 0.006** | 0.008*** |
| | | (0.003) | (0.003) | | (0.002) | (0.002) |
| Sleep hours, avg/night | | -0.037*** | -0.033** | | -0.002 | -0.002 |
| | | (0.010) | (0.010) | | (0.008) | (0.008) |
| Sufficient sleep | | -0.010 | -0.032 | | 0.013 | -0.026 |
| | | (0.026) | (0.026) | | (0.023) | (0.028) |
| Overweight/Obese | | -0.084** | 0.014 | | -0.089*** | 0.035 |
| | | (0.027) | (0.030) | | (0.026) | (0.036) |
| Screen time scale, hours/week | | -0.001† | -0.001† | | 0.000 | 0.000 |
| | | (0.001) | (0.001) | | (0.001) | (0.001) |
| Male | | -0.016 | -0.017 | | 0.001 | -0.01 |
| | | (0.024) | (0.020) | | (0.022) | (0.021) |
| Grade | | 0.036 | 0.052 | | 0.016 | 0.058* |
| | | (0.024) | (0.041) | | (0.018) | (0.027) |
| Age | | -0.074*** | -0.074*** | | -0.046** | -0.041** |
| | | (0.020) | (0.020) | | (0.016) | (0.016) |
| Parent highest education | | 0.032** | 0.034** | | 0.025** | 0.025** |
| | | (0.011) | (0.011) | | (0.009) | (0.009) |
| Nomination error flag | | -0.340*** | -0.300*** | | -0.101† | -0.118* |
| | | (0.085) | (0.084) | | (0.052) | (0.053) |
| Black NH [a] (Ref: White NH) | | -0.116* | -1.157*** | | - | - |
| | | (0.055) | (0.107) | | | |
| AsAm/PI NH [a] | | 0.212*** | -0.776*** | | - | - |
| | | (0.049) | (0.098) | | | |
| Other NH | | 0.116** | 0.348*** | | - | - |
| | | (0.038) | (0.105) | | | |
| Hispanic | | 0.091† | 0.013 | | - | - |
| | | (0.049) | (0.061) | | | |
| *Dyadic homophily terms* | | | | | | |
| Same, Male | | | 0.404*** | | | 0.251*** |
| | | | (0.036) | | | (0.037) |
| Same, White NH | | | 0.406** | | | - |
| | | | (0.131) | | | |
| Same, Black NH | | | 2.823*** | | | - |
| | | | (0.156) | | | |

*(Continued)*

**Table 2.** (Continued)

| | School 1 ("Sunshine High") | | | School 2 ("Jefferson High") | | |
|---|---|---|---|---|---|---|
| | **Puberty** | **+ Confounds** | **+ Dyadic terms** | **Puberty** | **+ Confounds** | **+ Dyadic terms** |
| Same, AsAm/PI NH | | | 2.018*** | | | - |
| | | | (0.122) | | | |
| Same, Other NH | | | -0.224† | | | - |
| | | | (0.123) | | | |
| Same, Hispanic | | | 0.743*** | | | - |
| | | | (0.059) | | | |
| Similarity, Pub (Phys Dev) | | | -0.006 | | | -0.024 |
| | | | (0.021) | | | (0.021) |
| Similarity, Pub (Relative Timing) | | | -0.040* | | | -0.008 |
| | | | (0.019) | | | (0.018) |
| Similarity, Physical activity | | | -0.024*** | | | -0.020*** |
| | | | (0.006) | | | (0.005) |
| Similarity, Sleep duration | | | -0.026† | | | -0.032* |
| | | | (0.016) | | | (0.014) |
| Same, Sufficient sleep | | | 0.071† | | | 0.090† |
| | | | (0.040) | | | (0.046) |
| Same, Overweight/Obese | | | 0.211*** | | | 0.215*** |
| | | | (0.044) | | | (0.053) |
| Same, 9th grade | | | - | | | 1.338*** |
| | | | | | | (0.074) |
| Same, 10th grade | | | 1.140*** | | | 1.236*** |
| | | | (0.081) | | | (0.044) |
| Same, 11th grade | | | 1.106*** | | | 1.345*** |
| | | | (0.042) | | | (0.050) |
| Same, 12th grade | | | 1.223*** | | | 1.395*** |
| | | | (0.086) | | | (0.087) |
| Akaike Inf. Crit. | 33,916 | 33,703 | 29,972 | 25,910 | 25,850 | 24,159 |
| Bayesian Inf. Crit. | 33,975 | 33,928 | 30,373 | 25,963 | 26,009 | 24,434 |

Note:

†p<0.1;

* p<0.05;

** p<0.01;

*** p<0.001

[a] GW = geometrically-weighted; ESP = edgewise shared partners; AsAm/PI = Asian American Pacific Islander; NH = Non-Hispanic.

coefficients is relatively small. For instance, exponentiating the coefficient for similar relative pubertal timing among students at Sunshine High is equivalent to a 1.04 times greater likelihood of those individuals being friends, and similar sleep duration at Jefferson equivalent to a 1.03 times greater likelihood of friendship, compared with the 1.5 times greater likelihood of friendship among two students of the same sex.

## Correlates of tie formation by sex

Table 3 ("Sunshine High") and Table 4 ("Jefferson High") report on correlates of tie formation by sex by decomposing the whole-school network into single-sex networks to give focus to

**Table 3. Characteristics associated with friendship formation, stratified by sex, school 1 ("Sunshine High").**

| | Whole network | Network decomposition models | | | |
|---|---|---|---|---|---|
| | | 1.Girls | 1.Girls +Menarche | 2.Boys | 3.Mixed Gender |
| | n = 1408 | n = 680 | n = 633 | n = 731 | n = 942 |
| Edges | -6.327*** | -5.153** | -4.002 † | -6.080*** | -6.611*** |
| | (0.900) | (1.928) | (2.166) | (1.608) | (1.388) |
| GW Degree (0.2) [a] | 1.494*** | 1.614*** | 1.404*** | 1.365*** | 0.548*** |
| | (0.171) | (0.158) | (0.151) | (0.150) | (0.121) |
| GW ESP (0.2) [a] | 1.669*** | 1.928*** | 1.929*** | 1.870*** | 1.481*** |
| | (0.032) | (0.065) | (0.071) | (0.060) | (0.049) |
| Pub: Physical dev | 0.017 | 0.053* | 0.049* | -0.007 | 0.011 |
| | (0.012) | (0.024) | (0.025) | (0.025) | (0.019) |
| Pub: Relative timing | 0.005 | -0.021 | -0.025 | 0.034 | 0.008 |
| | (0.011) | (0.024) | (0.026) | (0.022) | (0.018) |
| Pub: Age of menarche | - | - | -0.010 | - | - |
| | | | (0.019) | | |
| Physical activity | 0.011*** | 0.014* | 0.016* | 0.009 | 0.011* |
| | (0.003) | (0.007) | (0.007) | (0.006) | (0.005) |
| Sleep hours, avg/night | -0.033** | -0.04† | -0.031 | -0.008 | -0.027 |
| | (0.010) | (0.022) | (0.023) | (0.021) | (0.017) |
| Sleep enough | -0.032 | 0.081 | 0.055 | -0.035 | -0.009 |
| | (0.026) | (0.054) | (0.056) | (0.057) | (0.042) |
| Overweight/Obese | 0.014 | 0.117† | 0.081 | 0.002 | 0.044 |
| | (0.030) | (0.066) | (0.074) | (0.052) | (0.046) |
| Screen time | -0.001† | -0.001 | -0.001 | 0.000 | -0.002† |
| | (0.001) | (0.002) | (0.002) | (0.001) | (0.001) |
| Male | -0.017 | - | - | - | - |
| | (0.020) | | | | |
| Grade | 0.052 | 0.018 | -0.066 | 0.013 | 0.059 |
| | (0.041) | (0.087) | (0.098) | (0.072) | (0.063) |
| Age | -0.074*** | -0.114** | -0.097* | -0.043 | -0.061* |
| | (0.020) | (0.043) | (0.046) | (0.036) | (0.030) |
| Parent highest education | 0.034** | 0.051* | 0.064** | 0.023 | 0.034* |
| | (0.011) | (0.023) | (0.024) | (0.022) | (0.017) |
| Nomination error flag | -0.300*** | -0.345* | -0.251 † | -0.325† | -0.445** |
| | (0.084) | (0.153) | (0.147) | (0.172) | (0.137) |
| Black NH [a] (Ref: White NH) | -1.157*** | -1.166*** | -1.108*** | -1.710*** | -1.190*** |
| | (0.107) | (0.230) | (0.239) | (0.201) | (0.167) |
| AsAm/PI NH [a] | -0.776*** | -0.828*** | -0.702** | -0.908*** | -0.939*** |
| | (0.098) | (0.209) | (0.219) | (0.171) | (0.159) |
| Other NH | 0.348*** | 0.495* | 0.553* | 0.119 | 0.233 |
| | (0.105) | (0.228) | (0.240) | (0.182) | (0.166) |
| Hispanic | 0.013 | 0.045 | 0.100 | -0.121 | 0.015 |
| | (0.061) | (0.125) | (0.133) | (0.105) | (0.093) |
| *Dyadic Similarity terms* | | | | | |
| Same, Male | 0.404*** | - | - | - | - |
| | (0.036) | | | | |
| Same, White NH | 0.406** | 0.606* | 0.759** | 0.245 | 0.353† |
| | (0.131) | (0.261) | (0.275) | (0.224) | (0.204) |

(*Continued*)

**Table 3.** (Continued)

| | Whole network | Network decomposition models | | | |
|---|---|---|---|---|---|
| | | 1.Girls | 1.Girls +Menarche | 2.Boys | 3.Mixed Gender |
| | n = 1408 | n = 680 | n = 633 | n = 731 | n = 942 |
| Same, Black NH | 2.823*** | 2.940*** | 2.969*** | 3.399*** | 2.761*** |
| | (0.156) | (0.320) | (0.333) | (0.311) | (0.237) |
| Same, AsAm/PI NH | 2.018*** | 2.370*** | 2.269*** | 1.870*** | 2.282*** |
| | (0.122) | (0.259) | (0.271) | (0.212) | (0.197) |
| Same, Other NH | -0.224† | -0.345 | -0.399 | 0.098 | -0.085 |
| | (0.123) | (0.253) | (0.210) | (0.214) | (0.198) |
| Same, Hispanic | 0.743*** | 1.028*** | 1.003*** | 0.697*** | 0.888*** |
| | (0.059) | (0.121) | (0.128) | (0.103) | (0.094) |
| Similarity, Pub (Phys Dev) | -0.006 | 0.009 | 0.023 | -0.052 | 0.01 |
| | (0.021) | (0.038) | (0.041) | (0.040) | (0.031) |
| Similarity, Pub (Relative Timing) | -0.040* | -0.034 | -0.038 | -0.055 | -0.021 |
| | (0.019) | (0.037) | (0.040) | (0.035) | (0.029) |
| Similarity, Pub (Age of Menarche) | - | - | 0.008 | - | - |
| | | | (0.031) | | |
| Similarity, Physical activity | -0.024*** | -0.034** | -0.034** | -0.036*** | -0.033*** |
| | (0.006) | (0.012) | (0.012) | (0.010) | (0.009) |
| Similarity, Sleep duration | -0.026† | -0.004 | -0.001 | -0.063* | -0.05† |
| | (0.016) | (0.031) | (0.034) | (0.031) | (0.026) |
| Same, Sufficient sleep | 0.071† | 0.026 | 0.024 | 0.103 | 0.051 |
| | (0.040) | (0.072) | (0.079) | (0.076) | (0.061) |
| Same, Ovwght/Obese | 0.211*** | 0.356*** | 0.335** | 0.103** | 0.302*** |
| | (0.044) | (0.087) | (0.098) | (0.076) | (0.068) |
| Same, 10th grade | 1.140*** | 1.294*** | 1.226*** | 1.031*** | 1.258*** |
| | (0.081) | (0.162) | (0.175) | (0.143) | (0.129) |
| Same, 11th grade | 1.106*** | 1.358*** | 1.406*** | 1.054*** | 1.178*** |
| | (0.042) | (0.087) | (0.095) | (0.080) | (0.065) |
| Same, 12th grade | 1.223*** | 1.545*** | 1.788*** | 1.175*** | 1.223*** |
| | (0.086) | (0.183) | (0.211) | (0.152) | (0.134) |
| Akaike Inf. Crit. | 29,972 | 8,039 | 7,013 | 9,277 | 13,555 |
| Bayesian Inf. Crit. | 30,373 | 8,370 | 7,360 | 9,613 | 13,907 |

Notes:

†$p < 0.1$;

* $p < 0.05$;

** $p < 0.01$;

*** $p < 0.001$;

20% White; 54% male; Public; West, Suburb; Grades 10–12; gender and gender homophily terms not included in Models 2–5.

[a] GW = geometrically-weighted; ESP = edgewise shared partners; AsAm/PI = Asian American Pacific Islander; NH = Non-Hispanic.

friendships between same-sex students. The first column is identical to the final model in Table 2 to allow for ease of comparison. The table also includes two models for girls: one (1. Girls) that includes two measures of pubertal development (physical development, peer-relative timing), and an alternate model that allows for the inclusion of age of menarche (1. Girls+Menarche); this is followed by a subnetwork model that includes boys only (2.Boys). We

**Table 4. Factors associated with friendship formation, stratified by sex, school 2 ("Jefferson High").**

| | Whole network | Network decomposition models | | | |
|---|---|---|---|---|---|
| | | 1.Girls | 1.Girls +Menarche | 2.Boys | 3.Mixed Gender |
| | n = 762 | n = 352 | n = 343 | n = 411 | n = 465 |
| Edges | -6.567*** | -7.608*** | -8.110*** | -5.561*** | -6.631*** |
| | (0.563) | (1.390) | (1.474) | (1.213) | (0.682) |
| GW Degree (0.2) [a] | 0.610* | 2.584*** | 2.118*** | 1.171*** | 0.469† |
| | (0.290) | (0.439) | (0.360) | (0.253) | (0.251) |
| GW ESP (0.2) [a] | 1.536*** | 1.662*** | 1.659*** | 1.690*** | 1.470*** |
| | (0.036) | (0.072) | (0.074) | (0.065) | (0.040) |
| Pub: Physical dev | 0.013 | 0.009 | 0.004 | 0.010 | 0.009 |
| | (0.013) | (0.030) | (0.031) | (0.030) | (0.016) |
| Pub: Relative timing | -0.009 | -0.042 | -0.024 | -0.005 | -0.002 |
| | (0.010) | (0.027) | (0.029) | (0.020) | (0.013) |
| Pub: Age of menarche | - | - | 0.039† | - | - |
| | | | (0.022) | | |
| Physical activity | 0.008*** | 0.001 | 0.000 | 0.010* | 0.006* |
| | (0.002) | (0.007) | (0.007) | (0.005) | (0.003) |
| Sleep hours, avg/night | -0.002 | 0.018 | 0.019 | 0.003 | 0.004 |
| | (0.008) | (0.019) | (0.019) | (0.017) | (0.010) |
| Sleep sufficiency | -0.026 | -0.066 | -0.061 | -0.036 | -0.017 |
| | (0.028) | (0.056) | (0.058) | (0.058) | (0.034) |
| Overweight or Obese | 0.035 | 0.085 | 0.113 | 0.119* | 0.036 |
| | (0.036) | (0.070) | (0.073) | (0.057) | (0.041) |
| Screen time | 0.000 | 0.000 | 0.000 | -0.001 | 0.000 |
| | (0.001) | (0.002) | (0.002) | (0.001) | (0.001) |
| Male | -0.01 | - | - | - | - |
| | (0.021) | | | | |
| Grade | 0.058* | 0.203** | 0.214** | -0.043 | 0.078* |
| | (0.027) | (0.070) | (0.071) | (0.055) | (0.032) |
| Age | -0.041** | -0.109* | -0.126** | -0.009 | -0.048* |
| | (0.016) | (0.043) | (0.044) | (0.031) | (0.020) |
| Parent highest education | 0.025** | 0.045† | 0.030 | 0.010 | 0.034** |
| | (0.009) | (0.024) | (0.024) | (0.019) | (0.011) |
| Restricted nominations | -0.118* | -0.298* | -0.305* | -0.094 | -0.219** |
| | (0.053) | (0.140) | (0.139) | (0.114) | (0.072) |
| *Dyadic Similarity terms* | | | | | |
| Same, Male | 0.251*** | - | - | - | - |
| | (0.037) | | | | |
| Similarity, Pub (Phys dev) | -0.024 | 0.019 | 0.039 | -0.076 | -0.027 |
| | (0.021) | (0.047) | (0.048) | (0.047) | (0.024) |
| Similarity, Pub (Relative timing) | -0.008 | -0.033 | -0.033 | 0.018 | 0.000 |
| | (0.018) | (0.042) | (0.045) | (0.034) | (0.022) |
| Similarity, Pub (Age of menarche) | - | - | -0.068 † | - | - |
| | | | (0.035) | | |
| Similarity, Physical activity | -0.020*** | -0.017 | -0.016 | -0.039*** | -0.020** |
| | (0.005) | (0.013) | (0.013) | (0.009) | (0.006) |
| Similarity, Sleep duration | -0.032* | -0.056† | -0.059* | -0.061* | -0.041* |
| | (0.014) | (0.029) | (0.030) | (0.029) | (0.016) |

*(Continued)*

**Table 4.** (Continued)

| | Whole network | Network decomposition models | | | |
|---|---|---|---|---|---|
| | | **1.Girls** | **1.Girls +Menarche** | **2.Boys** | **3.Mixed Gender** |
| | **n = 762** | **n = 352** | **n = 343** | **n = 411** | **n = 465** |
| Same, Sufficient sleep | 0.090† | 0.187* | 0.182* | 0.109 | 0.077 |
| | (0.046) | (0.082) | (0.085) | (0.086) | (0.054) |
| Same, Overweight or Obese | 0.215*** | 0.351*** | 0.332*** | 0.287** | 0.198** |
| | (0.053) | (0.097) | (0.102) | (0.086) | (0.063) |
| Same, 9th grade | 1.338*** | 1.837*** | 1.775*** | 1.405*** | 1.409*** |
| | (0.074) | (0.186) | (0.186) | (0.144) | (0.089) |
| Same, 10th grade | 1.236*** | 1.728*** | 1.738*** | 1.334*** | 1.272*** |
| | (0.044) | (0.110) | (0.110) | (0.082) | (0.054) |
| Same, 11th grade | 1.345*** | 1.465*** | 1.449*** | 1.687*** | 1.350*** |
| | (0.050) | (0.124) | (0.126) | (0.104) | (0.059) |
| Same, 12th grade | 1.395*** | 1.341*** | 1.422*** | 1.905*** | 1.420*** |
| | (0.087) | (0.192) | (0.201) | (0.183) | (0.101) |
| Akaike Inf. Crit. | 24,159 | 5,662 | 5,419 | 7,355 | 17,572 |
| Bayesian Inf. Crit. | 24,434 | 5,879 | 5,652 | 7,579 | 17,817 |

Notes:

†p<0.1;

* p<0.05;

** p<0.01;

*** p<0.001;

97.3% White; Public; Midwest; Rural; Grades 9–12; Gender and gender homophily terms not included in Models 2–5.

[a] GW = geometrically-weighted; ESP = edgewise shared partners; AsAm/PI = Asian American Pacific Islander; NH = Non-Hispanic.

treat the final model (3.Mixed sex) as a residual category that omits same-sex ties. Comments below focus on the presence of sex variation.

At Sunshine High (Table 3), endogenous network terms are largely similar across subnetworks. Interestingly, although in the whole network there was no significant association between physical development and tie formation, examination of the network decomposed by sex revealed that at least some youth with advanced physical development are more likely to form friendships. More specifically, more physically developed girls tend to have more friendships (b = 0.053, p = 0.026). In addition, girls who are more physically active tend to form more friendships, with a coefficient size roughly the same (b = 0.014, p = 0.032) as the residual mixed-sex subnetwork (b = 0.016, p = 0.026); PA levels are not correlated with friendship among boys at this school. An inverse relationship among girls indicating older girls have fewer friendships (b = -0.11, p = 0.008) is not observed for boys. In terms of dyadic homophily, across all models, youth with similar levels of PA tend to be friends, while boys with similar amounts of sleep tend to form friendships (b = -0.063, p = 0.04).

At Jefferson High (Table 4), similar effects for endogenous network tendencies hold across subnetworks. Inspecting the menarche model, girls' ties were marginally more likely if they had a more advanced age of menarche (b = 0.039, p = 0.08) or in higher grades (b = 0.214, p = 0.003), though less likely if girls were older (b = -0.126, p = 0.004). Dyadically, friendships were more likely for girls with similar sleep duration (b = -0.059, p = 0.046), the same sleep sufficiency status (b = 0.182, p = 0.032), and marginally more likely for girls with similar age of menarche (b = -0.068, p = 0.055). Among boys, friendships were more likely for those who

were more physically active (b = 0.010, p = 0.04), or who were overweight/obese (b = 0.118, p = 0.036). Additionally, friendships were more likely among boys with similar PA levels (b = -0.039, p = <0.001) and similar sleep duration (b = -0.061, p = 0.036).

## Discussion

This study is the first to analyze how pubertal development, physical activity, and sleep–which are often studied separately–are independently correlated with friendship tie formation using a rigorous network analytic approach that accounts for statistical dependencies between participants and important contextual confounders. As expected from prior research using different methods, we conceptually replicated the finding that physical activity levels were correlated among youth who considered each other friends. Given that ERGMs share conceptual foundations with the stochastic actor-oriented model (SAOM) approach that Shoham and colleagues [37] and Simpkins et al. [13] used in these same two schools, this was not at all surprising. But we also replicated regression modeling-based trends towards youth with similar sleep duration [15] and, to a lesser extent, sleep sufficiency [40] selecting one another as friends with a more precise modeling technique that accounts for interpersonal statistical dependence. This makes sense intuitively, both in cases where friends are socializing in-person at night in ways that lead to a common shared bedtime, as well as cases where they are not together, given bedtime norms and the likelihood of youth engaging in similar evening behaviors, such as watching similar TV shows or talking on the phone. Taken together, these findings reinforce that a network perspective in sleep research continues to show promise.

We found qualified support for our first hypothesis, that adolescents' pubertal development would be associated with a greater likelihood of tie formation. In particular, we found that in the context of all relationships at Sunshine High (which had a high degree of SES and racial/ethnic diversity), youth with similar levels of peer-relative pubertal timing had a slight tendency to form more friendships with one another (although the effect size was small). Substantively, this means that if one student answered, "I look older than most", that student was more likely to be friends with another student who also reported more advanced development. Turning to the very different demographic composition of Jefferson High, we did not observe any relationships between pubertal development and social ties in the whole-network model. It is possible that visible markers of pubertal development "stick out" to peers more in school settings like Sunshine with a great deal of socioeconomic and racial/ethnic variation in the student population, making friendships on the basis of perceived relative pubertal timing more likely.

Next, we gained additional insight by stratifying network models by sex, in order to isolate how pubertal development and activity behaviors may be correlated within same-sex and opposite-sex relationships (to test Hypothesis 2). We found that in one of the schools (Jefferson), girls with similar age of menarche were nearly 7% more likely to be friends, though this association was only marginally significant. We did not observe homophily on physical development or peer-relative pubertal timing among girls, and there were no significant associations for any pubertal development measure among boys. In the larger, more diverse school (Sunshine), boys with similar amounts of sleep were more likely to be friends, and girls with more advanced physical development were more likely to have more friendships with other girls (regardless of those girls' pubertal development). Interpreting this association is complicated by the way the survey question was asked. As previously noted, respondents were prompted to note bodily changes related to pubertal development and gauge the extent of change since grade school. We then used multiple indicators of bodily change to construct a scale indexing very little development (1), to a great deal of development (5). For example, if a

girl's index responses indicated that she was "much more advanced" than she was in grade school, it could mean that she was, indeed, more physically developed (in an absolute sense), but it could also mean that she was a late developer (relative to peers). Unfortunately, survey responses given do not allow us to probe pubertal tempo.

At a minimum this association provides suggestive evidence that there is variation for girls between physical development and tie formation that we do not observe for boys. It may be that physical development is a more salient predictor of friendship for adolescent girls, with potential implications for health and well-being disparities. Early work (focused on girls) suggests that adolescents who develop earlier than their peers may select friends with similar physical development over chronological age [26, 72]. They may also spend more time with boys, which can be stressful for early developing girls who are not socially or cognitively prepared for the social stressors associated with mixed-sex peer interactions [73, 74]. These social patterns may put early developing girls at higher risk for poor developmental outcomes including aggressive and delinquent behaviors and adolescent dating abuse [28, 75–77]. On the other hand, it may also be the case that we did not observe these same associations between pubertal development and friendship for boys because boys develop later than girls. Analysis of an adolescent cohort with an average age two years older than ours may give more insight into this question. Our efforts build on the work of Kretsch and colleagues [20], who were the first to study homophily in pubertal development among socially-connected boys and girls using a regression-based approach. We extend their efforts with a more precise statistical social network approach able to analyze cross-gender relationships and health behavior confounders while accounting for statistical dependence between linked observations. Our work also comports more closely with the statistical network research of Franken and colleagues [19] who found that Dutch youth with similar pubertal development tended to be friends, rather than that of Lee and colleagues [24] who found no support for friendship based on pubertal development similarity among Taiwan youth. We would speculate that US-based youths' affinities with the Dutch case might owe more to youth socialization and friendship norms in Western societies rather than intercultural differences in pubertal timing, though more research is needed to assess this.

There are a host of modeling choices that warrant discussion in interpreting results. A perennial observational research concern is reciprocal causation (i.e. whether friendship changes cause behavioral shifts, or whether behaviors lead to friendship). Without more granular longitudinal information on behavior and friendship changes to establish temporal ordering, a cross-section of yearly data can provide insight into associations, but cannot provide causally informative evidence of statistical relationships. Participants' self-report behavior measures also reflect the era in which they were gathered (1994–95), prior to both the widespread dissemination of devices to measure PA and sleep, and prior to the emergence of portable devices with screens. While activity type and frequency are less precise than current standards, these measures have been widely scrutinized [49–51, 78] and are similar to those in U.S. surveillance data, such as NHANES [79]. Although we are limited in what we can infer from the self-reported measures of puberty (e.g. Add Health does not include physician-administered Tanner staging or hormone samples) and we have no information on pubertal tempo (which requires repeated puberty assessment starting in childhood), the pubertal development measures have adequate variation, and these measures have been examined in high-quality analyses exploring pubertal development [20, 27, 54].

Future research may productively extend our approach here to test how secular changes in screen use and changes in puberty timing since the mid-1990s may have both independent and interactive effects on contemporary friendship formation and maintenance. Given that the causes and consequences of how we interact on-line versus offline are not yet well-

understood [56], it is possible that connecting to friends in person in addition to online enriches relationships, though we can envision social connection being in some ways "thinner" and diminishing high-quality friendships in some ways. Separately but related, given that the age of puberty onset continues to decline, depending on whether it does so uniformly or not across biological sex and other population racial and ethnic subgroups, it is plausible that this leads to either no change in friendship stability, or more friendship turnover owing to widening gaps between physical and psychosocial maturity [59, 80].

As with all panel studies, it is not possible to adjust for unmeasured cofounders that occur between waves. However, by asking participants about their details of pubertal development during adolescence, Add Health improves upon early chronic disease research that previously relied on adults' retrospective reporting of puberty and health during their own youth [81–83]. In addition to the value that a network-based analytic framework that accounts for interpersonal dependencies offers, another key strength of the study was the use of three different measures of pubertal development (i.e., physical development and peer-relative timing standardized across boys and girls, age of menarche for girls), and a comparative framework across two very different schools (demographically as well as geographically). Yet it would be inappropriate to claim that our findings generalize to all schools. Given differences in school types, sizes, student composition, and cultures, care and thoughtfulness are warranted in applying these findings to school settings beyond the two types studied here. As McFarland and colleagues [43] point out in relational analyses of multiple school environments, giving attention to specific network ecologies of different schools is critical in considering how interventions might be implemented most successfully. Our study follows suit, demonstrating that it is important for the study of pubertal development to account for the network context of other physical health-promoting behaviors (PA, sleep) and risk factors (ST, weight status) that may independently shape, as well as confound, puberty's relationship with friendship formation. Future network research on puberty would benefit from more careful attention to pubertal timing and tempo, as well as how individuals of different sexes, genders, and ethnicities *think* about pubertal development differently. For instance, current studies that provide far more refined measurement of pubertal development and related neurobiological processes [84–86] could be augmented with high-quality dyadic friendship data [e.g., 87–89] to provide an opportunity to build upon what we have done here.

## Conclusions and practical applications

So what might we do with the insight that friendships are more likely between youth with similar relative pubertal timing, and possibly between girls with similar age at menarche? Although this study was conducted using data from youth in the mid-1990s, we believe the results can still inform current dialogues about child development and socialization. Despite differences in how peers socialize between then and now, many aspects of peer influence (e.g., social comparison, social desirability, popularity) transcend digital spaces, shaping young people's behaviors, attitudes, and self-perceptions. In particular, social media may encourage youth to compare their appearance to others even more than they did a few decades ago. There is a clear link between social media use and body dissatisfaction [90–92]. Additionally, exacerbated by AI-driven beauty filters, social media images can also reinforce biases (e.g., sexism, racism) and facilitate misinformation and deception [93]. At the same time, there is also some evidence that social media can provide a platform for body positivity and support [94]. Either way, the heightened attention to both positive and negative body messaging underscores how social media may exacerbate attention to physical features that change across development (e.g.,

skin/acne, body shape, hair growth). Indeed, our associational findings are likely to be conservative relative to today's youth.

Further, our research also offers important insights into other developmental contexts that remain highly relevant today. For example, we found differences by gender and school-based demographic composition. Specifically, pubertal timing appeared to be a more salient aspect of friendship selection for girls compared to boys, and in the school that had a high degree of SES and racial/ethnic diversity, versus a more homogenous school setting. These findings highlight the need for tailored interventions that not only take into account one's social identity, but also how one's identity maps onto the demographic composition of the school. Overall, our hope is that this study can provide a roadmap for one way to study the associations (and hopefully, prospective associations in a longitudinal framework) between physical development and friendships) across diverse contexts.

From one perspective, these findings complement the value of having conversations about physical and physiological changes during adolescence with trusted adults. For instance, further research could illuminate whether such discussions are more usefully structured not just in age or grade-matched groups (as is commonly done in school-based settings at present), but possibly in developmentally-matched groups, or further, within small groups of developmentally-matched friends. However, while it is possible that pubertal timing and tempo may be shifted at the population level through social ecological intervention (e.g., nutrition, obesity prevention), for the most part individuals cannot choose to change their pubertal development status, nor is this what we would advocate. Instead, this study reinforces that caregivers, teachers, and health professionals should continue to be educated about pubertal development, including our findings that physical development and timing may play a role in how youth socially connect with one another–even if these youth may not always be aware of phenotypic differences being at all responsible for friendships in question.

## Supporting information

**S1 File. Please see separate supporting information file for this manuscript, which describes data preparation and sample model code (Part 1) and goodness of fit model diagnostics (Part 2) in the same file.**
(DOCX)

## Acknowledgments

We appreciate input on earlier versions of this study from seminar participants at the Duke Network Analysis Center and the Dartmouth Interdisciplinary Network Research Group.

## Author Contributions

**Conceptualization:** Mark C. Pachucki, Lindsay Till Hoyt, John R. Sirard, Genevieve Chandler.

**Data curation:** Mark C. Pachucki.

**Formal analysis:** Mark C. Pachucki, Li Niu, Richard Carbonaro, Hsin Fei Tu.

**Funding acquisition:** Mark C. Pachucki, Lindsay Till Hoyt, John R. Sirard, Genevieve Chandler.

**Investigation:** Mark C. Pachucki, Lindsay Till Hoyt.

**Methodology:** Mark C. Pachucki, Lindsay Till Hoyt.

Project administration: Mark C. Pachucki, Lindsay Till Hoyt.

Resources: Mark C. Pachucki.

Supervision: Mark C. Pachucki, Lindsay Till Hoyt.

Writing – original draft: Mark C. Pachucki, Lindsay Till Hoyt.

Writing – review & editing: Mark C. Pachucki, Lindsay Till Hoyt, Li Niu, Richard Carbonaro, Hsin Fei Tu, John R. Sirard, Genevieve Chandler.

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
