## [Decision Letter · Decision Letter 0]

2 Jan 2024

PONE-D-23-37858Disentangling associations between pubertal development, healthy activity behaviors, and sex in adolescent social networksPLOS ONE

Dear Dr. Pachucki,

Thank you for submitting your manuscript to PLOS ONE. After careful consideration, we feel that it has merit but does not fully meet PLOS ONE’s publication criteria as it currently stands. Therefore, we invite you to submit a revised version of the manuscript that addresses the points raised during the review process.

Academic editor comments:Thank you for your well written manuscript. Agree with some of the concerns raised by Reviewers #1 and #3; would benefit from some more elaboration of limitations in the discussion section. 

We look forward to receiving your revised manuscript.

Kind regards,

Yasas Chandra Tanguturi

Academic Editor

PLOS ONE

 [Research reported in this study was supported by the National Institute of Nursing Research under Award Number R21NR017154 to Co-PIs Mark C. Pachucki and Lindsay T. Hoyt.].  

Reviewers' comments:

Reviewer's Responses to Questions

**Comments to the Author**

1. Is the manuscript technically sound, and do the data support the conclusions?

Reviewer #1: Yes

Reviewer #2: Partly

Reviewer #3: Yes

Reviewer #4: Yes

Reviewer #5: Yes

2. Has the statistical analysis been performed appropriately and rigorously? 

Reviewer #1: Yes

Reviewer #2: Yes

Reviewer #3: I Don't Know

Reviewer #4: I Don't Know

Reviewer #5: Yes

3. Have the authors made all data underlying the findings in their manuscript fully available?

Reviewer #1: No

Reviewer #2: No

Reviewer #3: Yes

Reviewer #4: No

Reviewer #5: Yes

4. Is the manuscript presented in an intelligible fashion and written in standard English?

Reviewer #1: Yes

Reviewer #2: Yes

Reviewer #3: Yes

Reviewer #4: Yes

Reviewer #5: Yes

5. Review Comments to the Author

Reviewer #1: This article presents a scientific approach to identifying key factors establishing friendships in pubertal age group adolescents. Network analysis posits an interesting approach to analyze this sociological phenomenon. The authors summarize a vast evidence base of literature on physical activity and sleep which lays good groundwork for the study. My review points are:

1. In the first line of the introduction, the authors may want to change the word "outsize".

2. Could the authors please explain how pubertal timing affected sleep activity in the Hoyt et al study, 2020.

3. The authors may want to expand on network analysis basics for the naive reader.

4. The study draws on data from the 1994-1995 study, a timeline when millenials were in the puberty. Given that we are in a different generation now, the authors may want to add this as a limitation for the generalizability of the study. Also, the use of devices have changed over time. Authors may wat to comment on this.

5. The assessment of pubertal development seems to be limited due to the tool used and the authors may want to allude to this as well .

6. The discussion may further expand on the utility of these findings in a greater way.

7. The authors may want to add a limitations section to ennumerate the limitations related to the date of the data collection and possibly add what steps may be taken to possibly replicate the study in todays pubertal adolescents in order to have some more impactful results that have further value.

If these recommendations are addressed, this paper may be suitable for publication but otherwise the paper possibly only has limited historical value and hence should be rejected.

Reviewer #2: The paper attempts to explore an interesting area of research regarding the interactions between pubertal development, healthy activity behaviors, and social networks among adolescents. A few points to improve the article.

The paper's extract lacks comprehensiveness by omitting the study's findings and conclusions. It focuses primarily on the abstract, introduction, and methods, which leaves a reader without a full understanding of the research's outcomes. There should be a different heading of Conclusions

Discuss the limitations of cross-sectional data in establishing cause-effect relationships. It is mentioned in one line only.

The absence of a discussion on the generalizability of findings diminishes the argument's broader relevance, especially when data is drawn from specific settings. Comment on this in the limitation as well.

Discuss the practical implications and real-world applications as a separate heading to enhance its relevance beyond academic circles.

Reviewer #3: Thank you for the opportunity to review the manuscript, “Disentangling associations between pubertal development, healthy activity behaviors, and sex in adolescent social networks.” The study is a detailed analysis of how pubertal timing/development, physical health, sleep, and biological sex influence friendship formation in adolescents. The paper is very well-written and provides sufficient detail on the study, methods, and findings. Overall, this is an interesting paper that contributes to the literature by using a novel statistical approach to the question of whether puberty, physical health, and more influence friendship formation. With these strengths in mind, a few concerns remain.

- Of most substantial concern is the timeline of the data collected. Specifically, as the authors highlight, the collected data is from 1994-1995. While the results and discussion presented are interesting, it is difficult to determine whether similar results would be found in today’s adolescent population. With the social landscape of 1994-1995 being quite different from today (e.g., prevalence of social media, online friendships, etc.), it would helpful for the authors to discuss how their findings may be translated to be relevant for adolescents in 2023. Similarly, the average age of pubertal onset is younger today than 20-30 years ago- how might an even earlier pubertal onset for some further influence friendship formation and social connections?

- The hypotheses are unclear. The entire “Hypotheses” section is quite long, and therefore, it would likely aid in reader comprehension to have a short, concise summary of the hypotheses at the end of the section to clearly define Hypothesis 1, 2, etc.

- Are the authors able to present a means of interpreting the size of an effect in the Results for readers unfamiliar with the statistical approach. Especially as there are instances when the authors cite a “marginal” effect for a p-value of 0.08 (which some would argue is not marginal/trend). As such, providing a means to easily understand the effect size, rather than relying on p-values, would be beneficial.

- In the Discussion, the authors conclude the manuscript discussing what to do with this information. It seems more time could be spent focusing on this in the Discussion, as currently, one might come away from reading the manuscript still questioning the impact and relevance of these findings. For this reviewer personally, while I walked away from the manuscript thinking the results were intriguing, I was still left with wondering ‘now what?’ or ‘so what?’.

Reviewer #4: This is a well-written manuscript. I liked reading this manuscript and believe that it is very promising. At the same time, I identified couple of issues that require the authors’ attention. Introduction is too long and same goes with the conclusion and discussion which can be combined and shortened to improve the readability of the article.

Professional use of English language is at par. Overall, this will be a good addition to current available literature.

Reviewer #5: Very interesting topic and study. It seems to be a topic that is difficult to study and have clear findings about given the number of confounders that are possible (e.g how close to they live, same route they take, where they sit in class, etc). Although there were not many associations found, there were a few that seem significant.

-What do you account for the increased likelihood of friendships in young with similar sleep duration?

-I don't see what were the age/grade ranges of the adolescents included in the ADD health study since you later mention perhaps older boys might show more association? Also, I'm sure grade level has a very strong association among friends. It might help to correct for grade? I'm not sure if that was done.

- How do you account for the differences between your findings and that of the Dutch study?

6. PLOS authors have the option to publish the peer review history of their article (what does this mean?). If published, this will include your full peer review and any attached files.

Reviewer #1: No

Reviewer #2: No

Reviewer #3: No

Reviewer #4: No

Reviewer #5: No

---

## [Author Response · Author response to Decision Letter 0]

31 Jan 2024

Dr. Yasas Chandra Tanguturi 

Editor: PLOS ONE

January 31, 2024

Dear Dr. Yasas Chandra Tanguturi,

We have completed revisions to our manuscript entitled “Disentangling associations between pubertal development, healthy activity behaviors, and sex in adolescent social networks” (PONE-D-23-37858) and we are pleased that a resubmission was invited for possible publication in PLOS ONE. We want to thank the editor and five reviewers for their close reads and constructive and thoughtful comments on our manuscript, and we think the manuscript is stronger as a result of considering their questions. Below we explain how editor (Part 1) and reviewers’ (Part 2) comments have been addressed in this version of the manuscript, with our responses in bold.

As you asked us to include an amended Role of Funder statement in this cover letter, please find the text directly below. Please note that our manuscript was supported by a single NIH grant mechanism (R21NR017154) to co-PIs Pachucki and Hoyt. However, the data collection for the parent study (going back to 1994) was supported by three other federal grant mechanisms. As a condition of data use by the data provider, any author that uses these data is required to acknowledge the funding mechanisms for the parent study as well. We tried to be very careful in how we acknowledged both (our direct funding, and the original parent study funding which we had nothing to do with). 

Research reported in this study was solely supported by the National Institute of Nursing Research under Award Number R21NR017154 to Co-PIs Mark C. Pachucki and Lindsay T. Hoyt. As a condition of data use, we acknowledge that this research uses data from the National Longitudinal Study of Adolescent to Adult Health (Add Health). Add Health is directed by Robert A. Hummer and funded by the National Institute on Aging cooperative agreements U01AG071448 (Hummer) and U01AG071450 (Allison E. Aiello and Hummer) at the University of North Carolina at Chapel Hill. Waves I-V data are from the Add Health Program Project, grant P01HD319121 (Harris) was directed by Kathleen Mullan Harris and designed by J. Richard Udry, Peter S. Bearman, and Kathleen Mullan Harris at the University of North Carolina at Chapel Hill, and funded by Grant P01HD31921 from the Eunice Kennedy Shriver National Institute of Child Health and Human Development, with cooperative funding from 23 other federal agencies and foundations. No direct support was received from grant P01HD31921, U01AG071450, or U01AG071448 for this analysis. The funders had no role in the current manuscript’s study design, analysis, decision to publish, or preparation of the manuscript.

Sincerely,

Mark Pachucki

o/b/o authors

[for details please see reference Response to Reviewers.pdf]

---

## [Decision Letter · Decision Letter 1]

5 Mar 2024

Disentangling associations between pubertal development, healthy activity behaviors, and sex in adolescent social networks

PONE-D-23-37858R1

Dear Dr. Pachucki,

We’re pleased to inform you that your manuscript has been judged scientifically suitable for publication and will be formally accepted for publication once it meets all outstanding technical requirements.

Kind regards,

Yasas Chandra Tanguturi

Academic Editor

PLOS ONE

Additional Editor Comments (optional):

Reviewers' comments:

Reviewer's Responses to Questions

**Comments to the Author**

1. If the authors have adequately addressed your comments raised in a previous round of review and you feel that this manuscript is now acceptable for publication, you may indicate that here to bypass the “Comments to the Author” section, enter your conflict of interest statement in the “Confidential to Editor” section, and submit your "Accept" recommendation.

Reviewer #1: All comments have been addressed

Reviewer #3: All comments have been addressed

Reviewer #4: All comments have been addressed

Reviewer #5: All comments have been addressed

2. Is the manuscript technically sound, and do the data support the conclusions?

Reviewer #1: Yes

Reviewer #3: Yes

Reviewer #4: Yes

Reviewer #5: Yes

3. Has the statistical analysis been performed appropriately and rigorously? 

Reviewer #1: Yes

Reviewer #3: I Don't Know

Reviewer #4: I Don't Know

Reviewer #5: Yes

4. Have the authors made all data underlying the findings in their manuscript fully available?

Reviewer #1: Yes

Reviewer #3: Yes

Reviewer #4: No

Reviewer #5: Yes

5. Is the manuscript presented in an intelligible fashion and written in standard English?

Reviewer #1: Yes

Reviewer #3: Yes

Reviewer #4: Yes

Reviewer #5: Yes

6. Review Comments to the Author

Reviewer #1: (No Response)

Reviewer #3: The authors have done an excellent job of thoughtfully and thoroughly responding to reviewer concerns. I have no additional comments.

Reviewer #4: All the comments haves been addressed by the author. Revised manuscript seems better than original manuscript.

Reviewer #5: Thanks for the responses to the reviewer's questions and incorporating it into the manuscript. I think the write up overall looks better.

7. PLOS authors have the option to publish the peer review history of their article (what does this mean?). If published, this will include your full peer review and any attached files.

Reviewer #1: No

Reviewer #3: No

Reviewer #4: No

Reviewer #5: No

---

## [Editor Report · Acceptance letter]

4 May 2024

PONE-D-23-37858R1 

PLOS ONE

Dear Dr. Pachucki, 

I'm pleased to inform you that your manuscript has been deemed suitable for publication in PLOS ONE. Congratulations! Your manuscript is now being handed over to our production team.

Kind regards, 

on behalf of

Dr. Yasas Chandra Tanguturi 

Academic Editor

PLOS ONE